# Influence of Sociodemographic, Organizational, and Social Factors on Turnover Consideration Among Eldercare Workers: A Quantitative Survey

**DOI:** 10.3390/ijerph20166612

**Published:** 2023-08-20

**Authors:** Sabina Krsnik, Karmen Erjavec

**Affiliations:** Faculty of Health Sciences, University of Novo Mesto, Na Loko 2, 8000 Novo Mesto, Slovenia; karmen.erjavec@uni-nm.si

**Keywords:** long-term care, elderly care, staff turnover, social recognition, working environment

## Abstract

Staff turnover in long-term care (LTC) is considered one of the main causes of staff shortages and a key problem for LTC systems in the developed world. Factors affecting staff turnover in LTC facilities are poorly understood due to a fragmented approach. The aim of this study was to use multivariate analysis to identify the factors at the macro-, meso-, and micro-level that influence LTC workers’ turnover in Slovenia, a typical Central and Eastern European country. A correlational cross-sectional survey design with a self-reported online questionnaire was used among Slovenian LTC workers (*N* = 452). The results show that more than half of LTC workers intend to quit their jobs and leave the LTC sector. LTC workers who intend to leave are generally younger, have worked in the LTC sector for a shorter period, are mainly employed in the public sector, especially in nursing homes, and earn less. The connection between the intention to leave and the factors at the macro-, meso-, and micro-level is very high. Over 75% of the variance of intention to leave was explained by the linear influence of sociodemographic characteristics, social recognition, and work environment. Urgent measures for improving the work environment are needed.

## 1. Introduction

High turnover rates combined with a great need for long-term care (LTC) due to an aging population have led to a shortage of LTC staff in most developed countries [1,2,3,4,5,6,7,8]. The COVID-19 pandemic, which disproportionately affected the LTC sector, has made an already difficult situation for LTC workers—the high workload and the large number of increasingly seriously ill residents—even worse [9,10]. For example, nursing homes, which are primarily social institutions, have been transformed into hospitals for elderly patients with COVID-19 in Slovenia, other Central and Eastern European countries, as well as many countries around the world. Staff turnover in LTC facilities increased between 10 to 20% during the pandemic; for example, in the United Kingdom, the pre-pandemic turnover rate was 20%, and in 2021 over 36% of the nursing workforce reported their intention to leave LTC due to increased pressures at work [11]. As a result of additional staff turnover during the COVID-19 pandemic, there are now even fewer LTC workers, and hiring new employees is even less appealing. Despite the extraordinary need for LTC and lack of housing, Slovenia, a typical Central and Eastern European country with a rapidly growing elderly population, has a shortage of about 25% for LTC workers [12], leaving about 10% of beds in LTC settings unoccupied [13]. Nursing shortages negatively impact the quality of care [14] and overall LTC efficiency [1], resident/patient mortality [15], as well as higher recruitment and replacement costs and training costs for new staff. Additionally, due to under-staffing and overburdening of the remaining LTC workers, health and safety risks for employees increase and, in the worst-case scenario, result in accidents with serious human costs [16]. Thus, due to the increasing labor shortage, the strain on the remaining staff also affects the workers themselves, leading many workers to further leave the LTC sector [11,16]. Therefore, the complex underlying mechanisms that lead LTC workers to quit their jobs should be considered, as we still do not know exactly why and how they leave their jobs.

To better understand LTC staff turnover, the intention of employees to leave their jobs in the LTC sector should be discussed. Previous studies have shown that turnover intention is an important, practical predictor variable of turnover and the strongest predictor of actual turnover behavior [17].

Existing studies indicate that many factors influence staff turnover in the LTC sector [18,19,20,21,22,23,24,25]. First, macro-level factors such as economic growth, unemployment rate, and comparable income affect LTC staff turnover [18]. Indeed, LTC workers are among the lowest-paid workers in Slovenia and in the EU; their wages are comparatively lower than in the health sector, leading to a migration of LTC workers to the competing health sector. Weak collective bargaining and union representation also contribute to the poor situation of employees [18]. There are other factors related to social macro trends that influence LCT worker turnover, such as exploitative migration practices, gender inequality, and inadequate funding of the sector [18]. The perception of misrecognition of LTC workers also plays an important role at the societal level, as society (including authorities, the media, and the public) does not see, hear, value, or consider them [19,20]. In general, working in LTC is seen as low-status and poorly respected [20]. A survey among Nordic eldercare workers showed that 41% had seriously considered quitting, a third felt unappreciated by the top municipal leaders, and a quarter felt unappreciated by the mass media. The study also found that perceived misrecognition has a significant and independent influence on LTC workers’ concerns about leaving their jobs [21]. A recent Slovenian qualitative study has shown that the lack of geriatric content in secondary and tertiary education is also relevant to the work of LTC workers, as candidates for work in LTC are not adequately prepared to work with older people [22].

Second, many studies have almost unanimously mentioned organizational-level factors such as wages, working conditions, training, career development, and occupational health and safety [1,2,3,4,5,6,7,18,19,20,21,22,23]. Poor wages and working conditions have been identified as the primary factors contributing to high staff turnover in this sector. The working conditions of LTC workers can vary depending on the country, the specific care setting, and employer practices; however, there are some common aspects, such as LTC workers often facing high workloads due to caring for people with complex needs, shift work, physically demanding tasks, and emotional and psychological stress [1,2,3,4,5,6,7,18]. According to a national survey of Norwegian nurses in 2018, 25% wanted to work outside LTC facilities, and working conditions in LTC facilities were a significant predictor of the desire to leave [3]. A recent study showed that the working environment, including job demands (illegitimate work tasks, role conflicts, and role overload), resources for LTC workers (social support, job autonomy, investment in development, and work being meaningful), and the care process (consideration of patients’ beliefs and values, shared decision-making, engaging authentically, sympathetic presence, and providing holistic care) influence work engagement [7], which may affect staff turnover [23]. However, there are other important factors that could have an impact on staff turnover, such as work-life balance, which is a common challenge for LTC workers because they have demanding work and often work in shifts, meaning they need to balance their professional duties with their private and family lives [18]. A Slovenian qualitative study [22] has shown that, in contrast to other studies already mentioned [1,2,3,4,5,6,7,18,19,20,21,22,23], the flexibility of work and opportunities for rewards from managers, such as wage supplements, awards, and honors, as well as the possibility of on-the-job training, are particularly emphasized at the level of job resources, demands, role overload, and workload. This is a result of the fact that the norms defining the relationship between staff and residents in Slovenian LTC facilities have not changed for decades, even though the majority of residents have become seriously ill patients. At this level of the care process, the quality of care and the relationship with the patients and residents were highlighted, as they reported that they could not provide care at a high enough level due to the lack of staff but that they stayed in LTC because of a good relationship with the patients and residents.

Third, the interactions between employers, residents/patients/relatives, and nurses are also key contributors to employee turnover at the micro-staff level [21,24,25]. Turnover rises when nurses are not properly supervised and do not feel a sense of belonging with their employers [21]. Previous studies have found limited empirical support for a relationship between sociodemographic factors and LTC staff turnover. Some concluded that there is no association [4], others that younger caregivers are more likely to leave LTC facilities than older ones [3], and some that poor health status influences staff turnover [21]. As there are contradictory results on the sociodemographic characteristics of LTC workers who want to leave their jobs, this should be investigated.

Factors affecting staff turnover in LTC facilities are poorly understood due to a fragmented approach. Therefore, the question arises as to what factors influence LTC workers’ decisions to leave their jobs. The aim of this study was to use multivariate analysis to identify the most important factors at the macro-, meso-, and micro-level that influence LTC workers’ turnover. The specific objectives were to identify the proportion and sociodemographic characteristics of workers who intend to leave their jobs in LTC facilities, the area of search for a new job, and the macro-, meso- and micro-level factors that influence the turnover of workers in LTC facilities. This study contributes to filling the knowledge gap by analyzing the complex set of factors and considerations associated with employment termination in LTC facilities. We hypothesize that socioeconomic factors, social recognition, and work environment play important roles. Figure 1 illustrates the hypothesis. These findings may help policymakers apply the right strategies for recruiting and retaining employees in the Central and Eastern European LTC sectors.

## 2. Materials and Methods

### 2.1. Study Design

A cross-sectional correlation study was conducted. 

### 2.2. Study Setting 

A standardized self-reported online questionnaire was conducted between 13 March and 13 April 2023.

### 2.3. Study Size and Sampling Strategy

All Slovenian LTC facilities (59 public institutions and 44 nursing homes with a concession, 38 special social welfare institutions and occupational activity centers, 78 social centers, and 54 private contractors) were included in the study.

### 2.4. Measures and Instruments

An online, self-administered questionnaire consisted of three sets of independent variables, i.e., sociodemographic variables, psychosocial work environment, and social recognition, and one dependent variable, namely the intention to leave work.

*Sociodemographic variables* included age, education, income, role in the institution, organizational form of the institution, type of LTC organization, shift work, health status, and dependent household members.

Based on previous qualitative research findings [22], we have further developed a standardized psychosocial work environment scale, KIWEST, a commonly used instrument developed by a Norwegian university that has proven to be valid and reliable [26], to examine the impact of meso-level or organizational factors on staff turnover. *Job resources* included in the study were technical support, wage supplements, awards and honors, flexibility of work, stability of employment, possibility of workplace training, investment in employee development, management’s support, and job autonomy. *Job demands* included role overload, workload, and interpersonal conflicts. *Care process* included quality of care and relationship with patients/residents. *Work-life balance* was also included to identify outcomes of the work process. The items were rated on a five-point Likert scale ranging from 1—strongly disagree to 5—strongly agree. 

To investigate the impact of social recognition on staff turnover, a question was asked about recognition at the social level, i.e., at the level of employees and LTC work. Judging from the in-depth interviews, the (lack of) inclusion of geriatric content in secondary and tertiary education is very important for LTC workers.

The intention to leave work in LTC facilities was first assessed with one item: “Do you have the intention to leave your job in the next 6 months?”. Possible answers were “No,” “Uncertain,” and “Yes”. To the question “In which area would you like to find a job?” the following answers were provided: “I will stay in the LTC sector.”, “I will find a job in other health sector institutions.”, “I will find a job outside the LTC and health sector.”, and “Uncertain”. The calculated reliability of the measurement showed that the items had satisfactory discriminatory power, as the Cronbach reliability coefficient α was above 0.7 for all constructs.

A pilot study was conducted with LTC experts and LTC workers, and the questionnaire was improved where their suggestions were relevant.

### 2.5. Data Collection

For data collection, a survey technique was used, which allowed us to obtain data on the attitudes and opinions of the respondents. It also ensured the comparability of data between respondents, increased the speed and accuracy of data collection, and facilitated their processing. The invitation to participate in an online survey was sent to all Slovenian LTC facilities via publicly accessible e-mail addresses. The managers were asked to provide a link to the online questionnaire to all LTC workers in their institution. In the introductory section of the online questionnaire, it was indicated that completing and submitting the questionnaire was considered informed consent and voluntary participation. A total of 452 respondents completed the questionnaires in full, which corresponds to a representative sample [10]. 

### 2.6. Ethical Consideration 

The study protocol was reviewed and approved by the Ethical Committee of Human Research at the Faculty of Health Sciences, University of Novo Mesto, working within the Committee for Research (code number 8/2022). The Helsinki Declaration of the World Medical Association on the ethical principles for medical research involving human subjects was respected. 

### 2.7. Data Analysis

To determine the normality of data, the Shapiro-Wilk Test was used. The reliability of the measurement was checked by calculating the Cronbach reliability coefficient α.

To define the underlying dimensions of intention to leave work in the field of LTC, we used the principal component approach. The goal was to reduce the size of the variables to an actual underlying dimensionality. Barlett’s test for sphericity (*p* < 0.05) and the KMO statistic (>0.5) indicated that the analysis was reasonable. Based on the results of the component matrix, we defined 12 underlying dimensions of intention to leave work in the field of LTC related to the work environment, namely job autonomy, social recognition of workers, the meaningfulness of work, investment in employee development, investment in unit development (job resources), illegitimate work tasks, role conflicts, role overload, interpersonal conflicts (job demands), relationship with patients/residents, shared information (care process), and work-life balance. 

To determine if there were differences in intention to leave according to sociodemographic characteristics, the Kruskal-Wallis H test was used, which enabled us to determine whether the medians of two or more groups were different.

To test the hypothesis, each component thus defined was used in a linear regression analysis together with other independent variables to determine the relationship between each component and the intention to leave work. The main goal of linear regression analysis was to show the relative importance of predictors and estimate the effect of the independent variables on the dependent variable, namely the intention to leave work.

## 3. Results

### 3.1. Descriptive Data

The survey was completed by 452 respondents (Table 1), of whom 90.2% were female (9.8% male). Most respondents (32.1%) were between 41 and 50 years old, followed by those between 51 and 60 years (26.5%) and between 31 and 40 years (23.2%). Up to 32.5% of the respondents had completed secondary school, 26.8% had a college degree, and 20.4% had a university degree. Most respondents (24.2%) had between 11 and 20 years of service in the LTC sector, followed by those with 2 to 5 years (22.4%). As public LTC is predominant in Slovenia, 85% of workers worked in a public LTC organization. Most respondents worked in nursing homes (39.8%), followed by those who worked in centers for social work (20.8%) and special social welfare institutions (20.6%). Most respondents were members of the nursing/care team (45.6%) and worked a single shift (40.5%), a double shift including weekends and holidays (27.7%), or three shifts including weekends and holidays (17.9%). Most respondents received the minimum Slovenian wage and up to EUR 1100 net per month (25.1%), followed by those who received between EUR 1101 and 1300 net per month. Most care for a household member (67.5%). Over 70% did not have chronic diseases.

### 3.2. Intention to Leave and Area for Finding a New Job

More than half (55.1%) of the respondents had seriously thought about leaving their job in the last year. When those who intend to leave their job (52.4 %) in the next 6 months were asked in which area they would like to find a job, most respondents (68.8%) replied, “Uncertain”. Only 4.6% of respondents who planned to leave their job in the next six months would stay in the LTC sector; 6.6% wished to stay in the healthcare sector, and more than 20% planned to find a job outside the LTC and health sector (Figure 2).

The Kruskal-Wallis test showed statistically significant differences (*p* > 0.5) in seven sociodemographic characteristics of workers regarding their intention to leave their work in LTC (Table 2). Results show that workers who have the intention to leave in the next six months are, in general, younger. The highest share (32.2%) of workers who intend to leave were aged between 31 and 40, followed by workers aged between 21 and 30 (29.7%) and those aged between 41 and 50 (28.4%). Similarly, workers who had been working in the LTC sector for a shorter period of time would rather leave their work than those who had worked in this sector for more than 20 years. Namely, the majority of workers who had the intention to leave work in the LTC sector have worked there between 2 to 5 years (31.9%), followed by those who have 6 to 10 years of service (23.6%) and those who have worked for less than 2 years (18.1%). 

Based on the organizational form of the institution, workers who had the intention to leave were mainly employed in the public sector (73.0%), especially in a nursing home (63.5%), followed by specialized social welfare institutions (14.9%) and centers for social work (12.2%). The fewest workers that had the intention to leave were in centers for training, work, and care (1.4%), home care assistant institutions (2.7%), and occupational activity centers (5.4%).

Intention to leave work in the LTC sector also differed by the height of income, namely the monthly net wage. Results show that workers who earned less than EUR 1301.00 presented the highest share of those who intend to leave (78.4%). Related to income, 73.0% of workers who intended to leave had a household member or members who depended on them.

The highest share of workers who intended to leave thinks that the general public, politicians, and the media do not value their work (70.3%).

### 3.3. Factors Influencing Worker Turnover in LTC Facilities

To test the model presented in Figure 1, the multiple linear regression method was used. The following variables were included in the model: *sociodemographic characteristics* (age, education, years of service, income, shift work, dependent household members, health status, organizational form of the institution, and role in institution); *social recognition and geriatric education* (social recognition of LTC workers, social recognition of LTC work, and geriatric secondary and higher education); *job resources* (technical support, wage supplements, awards and honors, flexibility of work, stability of employment, possibility of workplace training, management’s support, job autonomy, meaningfulness of work, investment in employee development, and investment in unit development); *job demands* (role overload, workload, interpersonal conflicts, and illegitimate work tasks); *care process* (quality of care and relationship with patients/residents); and *work-life balance*.

In the calculation, we used the backward method and excluded statistically uncharacteristic variables (*p* < 0.05), namely age, years of service, role in institution, meaningfulness of work, investment in unit development, illegitimate work tasks, and shared information; thus, they had no effect on the intention to leave work in the LTC sector.

The calculation showed that the multiple correlation coefficient was 0.928, i.e., the connection between the intention to leave and the included dimensions is very high. The multiple coefficient of determination was 0.755, i.e., 75.5% of the variance of intention to leave was explained by the linear influence of sociodemographic characteristics, social recognition, geriatric education, and work environment. The F-test value was 8.051, and because the specificity level was less than 0.05 (*p* =< 0.000), we rejected the null hypothesis.

Table 3 shows that the following sociodemographic factors have a statistically significant influence on the intention to leave work in the LTC sector: education, income, shift work, dependence of household members, health status, and organizational form of the institution. Income has the greatest influence on the intention to leave (0.641), which means that if wages continue to be low, the intention to leave will continue to increase. The intention to leave will also increase the higher the education level of workers, with the need to work multiple shifts, with a higher number of those without a dependent member in their household, and for those who do not have chronic illnesses, which enables them greater opportunities.

At the social level, the results show the statistically significant influence of the social recognition of LTC work in long-term care and workers, as well as geriatric secondary and higher education, on the intention to leave. Geriatric education has the greatest positive influence on the intention to leave (1.442). This means that until workers are adequately prepared to work with older people during their formal education, the intention to leave will continue to increase. The results also show that as the number of workers who think that they and the work in LTC are not recognized by society increases, the intention to leave will increase. 

At the organizational level or the working environment, job resources (technical support, wage supplements, awards and honors, flexibility of work, stability of employment, possibility of workplace training, investment in employee development, management’s support, and job autonomy), job demands (role overload, workload, and interpersonal conflicts), the care process (quality of care and shared decision-making with residents/patients), and a negative work-life balance have statistically significant impacts on the intention to leave LTC. Among these, the following factors have the greatest influences on intention to leave: stability of employment (−1.965), relationship with patients/residents (1.889), the possibility of workplace training (1.595), workload (1.337), wage supplement (−1.204), award and honors (−1.279), management’s support (−1.181), and quality care (1.144). More LTC workers will leave the LTC sector if they feel that their job is less secure, that they do not have enough opportunities for additional training, that they have poorer relationships with patients or residents, that they are overworked and do not have management’s support, that they are not adequately compensated, i.e., overtime is not paid, that they do not receive awards or honors, or that they believe that the quality of long-term care in the organization is poor. Moreover, the intention to leave will increase if workers will feel that they have too great a workload and/or if they do not have a good balance between their work and family life.

## 4. Discussion 

### 4.1. Intention to Leave and Area for Finding a New Job

Using a nationwide sample in Slovenia, we investigated the socially significant challenge of the shortage of workers in LTC due to staff turnover and the increasing age of the population. In line with the objective of the study, it was found that only about 5% of respondents indicated that they would keep working in the LTC sector, while more than half (53%) of respondents had serious intentions to quit their jobs in the upcoming six months. A comparison with the results of existing studies showed that the proportion of LTC workers who intend to leave their jobs increased in European and developed countries during the COVID-19 pandemic [11,21] but did not come close to the identified proportion. This further demonstrates the severity of the staff turnover issue in Slovenia and other comparable Central and Eastern European countries. There could be several reasons contributing to higher staff turnover in LTC in Slovenia compared to other European countries. One possible reason could be the different economic development, which could lead to a higher level of job insecurity [27]. Another reason could be the different labor market conditions and employment opportunities. Western Europe, which is more developed and has a stronger economy, may offer a wider range of employment opportunities and better working conditions [27] that may attract and retain workers and lead to lower staff turnover. In addition, the availability of social support systems, such as unemployment benefits and social assistance programs [27], may differ between the two regions and influence workers’ decisions to stay in LTC. Results indicate that most Northern and Western European countries use public funds more effectively than Central, Eastern, and Southern European countries, which may be due to different approaches to social policy [28].

The results of the objective study on the sociodemographic characteristics of respondents who seriously intended to leave their jobs in the next six months show that workers who had the intention to leave in the next six months were, in general, younger, had worked in the LTC sector for a shorter period of time, were mainly employed in the public sector, especially in nursing homes, and earned less than EUR 1301.00. This is in line with a study that identified younger elderly caregivers as more likely to leave the LTC sector than older ones [3]. This could be explained by the fact that younger people with a higher education level often have higher expectations and ambitions due to their educational background, which has exposed them to different career opportunities and encouraged them to strive for career advancement. Not to mention their greater enthusiasm and eagerness compared to older LTC workers, who may have more practical and realistic expectations and prioritize stability and job satisfaction over career advancement due to their lengthy experience in the sector [29]. In addition, work in nursing homes tends to be physically demanding, emotionally exhausting, and monotonous compared to other LTC jobs [29,30]. The results also support the fact that LTC workers receive low pay. More than 70% of respondents earn less than the minimum wage in Slovenia (EUR 878/month in February 2023) and more than a third less than the average Slovenian net wage (EUR 1423/month in February 2023).

### 4.2. Factors Influencing Worker Turnover in LTC Facilities 

The main objective of the study was to use multivariate analysis to identify the macro-, meso-, and micro-level factors that influence the intention of LTC workers to leave. The results confirm the hypothesis that sociodemographic factors (education, income, shift work, dependence of household members, health status, and organizational form of the institution) have a statistically significant influence on the intention to leave work in LTC. Not surprisingly, income has the greatest influence on the intention to leave due to—as has already been explained—lower salaries in this sector. Thus, workers with a lower monthly income, higher education level, who do not have family members to care for, who work multiple shifts, who are involved in public care, and who do not have chronic illnesses are more likely to leave LTC. This could be due to the fact that workers with lower monthly incomes typically face a financial burden that makes it difficult for them to face the long hours and responsibilities associated with the job. Higher education levels usually lead to better employment opportunities outside of elder care that offer more attractive salaries, benefits, and opportunities for advancement; working multiple shifts can be physically and emotionally demanding for workers leading to burnout and a desire to seek a job with less strenuous demands. Similarly, people who work in public care may want less stressful jobs, as staff shortages are more common in public LTC because there are fewer opportunities to reward workers. Finally, people without chronic illnesses may have more physical and mental energy to look for other job opportunities [18,22].

At the societal level, the results confirm that social recognition and geriatric education have a statistically significant influence on the intention to leave work in LTC. The results show that most respondents think that society (social authorities, the media, and the general public) does not value their work in LTC, in contrast to their family members and close friends who highly appreciate and value their work. The effect of perceived social recognition on the intention of Slovenian LTC workers to leave is like that of Nordic LTC workers [21], which can be explained by the similarity of social values and cultural norms despite different origins. In Nordic countries, the emphasis on social recognition and equality is deeply embedded in the culture. Workers in these countries value recognition and appreciation for their contribution to society and seek validation from their colleagues and supervisors. If they feel misrecognized or undervalued, this can lead to increased dissatisfaction and a greater willingness to leave their jobs. On the other hand, the importance of social recognition in Slovenia and Eastern European countries is greater because of the socialist tradition [31]. 

Furthermore, there is a widespread reason for the social misrecognition of LTC workers. Workers in the LTC sector feel less recognized on a social level compared to their family members and close friends when their work is portrayed negatively in the media, which often focuses more on sensational stories, making LTC workers feel overlooked and undervalued [21,32,33]. Further, the results show that a lack of geriatric education in secondary and tertiary education has an important influence on the intention to leave. This can be explained by the fact that a lack of geriatric education can lead to a lack of understanding of the needs of the elderly population. Without this understanding, care professionals may feel overwhelmed or inadequately prepared when caring for older people [22]. For those who are not aware of the physical and cognitive challenges that older people may face, the probability of leaving the profession due to a lack of understanding of the needs of the older population is higher [7].

In addition to the macro- and micro-factors, the meso-factors, i.e., the organization or the work environment, also have an important influence on the intention to leave. Results confirm the hypothesis that the work environment has a statistically significant influence on the intention to leave work in LTC. This study shows that LTC workers are more likely to stay in the LTC sector if they feel secure, have more opportunities for training, have good relationships with patients or residents, do not feel overworked, believe that the quality of LTC in the organization is high, and have greater support from management, including adequate compensation for their work and contribution, i.e., paid overtime and with awards or honors. 

Why do these factors have the greatest influence on the intention to leave a Slovenian eldercare facility? Job stability is a key factor in Slovenia due to the small number of jobs on the market [22], as job security provides a sense of confidence and reliability that is necessary for long-term employment [34]. Relationships with patients/residents enable positive interactions that can lead to greater job satisfaction [35], which can lead to lower turnover. The possibility of workplace training keeps employees informed of their job duties and can foster a more positive work environment [36]; too much work can lead to burnout and job dissatisfaction [10]. Wage supplements, awards, and honors can all act as incentives to stay in a job, as recognition for a job well done can be rewarding [37]. Management’s support is key, as having a supportive team behind you can make a job much more enjoyable [7]. Finally, quality care—including a variety of factors such as the qualifications of the staff, the availability of resources, the overall cleanliness and safety of the premises, and the quality of the medical and nursing care—is essential, as providing quality care to patients/residents is an important part of the job and an important requirement for a person’s well-being in LTC facilities; without it, the individual’s health will suffer [38]. Poor quality care can lead to substandard care, which in turn can lead to medical complications, poor mental health, and a greater risk of infection [14].

On the other hand, the intention to leave is higher when workers feel they have too great a workload and if they do not have a good balance between their work and family life. In accordance with previous studies [1,2,3,4,5,6,7,18,19,20,21,22,23], where the work environment played a crucial role in determining job satisfaction, this study showed it is a key factor influencing workers’ intention to leave. Poor working conditions, including job demands, resources for LTC workers, and the care process, influence work engagement leading to increased stress and burnout among LTC staff, which affects their decision to leave the organization [7]. These findings support the Job Demand-Resource model [39,40], which assumes that job resources and job demands are the two main job categories that influence work engagement (motivation) and related organizational outcomes. Our research has extended this approach by showing that they influence the decision to leave. In addition, sociodemographic and social factors were also found to play a role, as shown in Figure 1. 

### 4.3. Implications

Based on the results, changes to the eldercare policy and measures for improving the work environment are urgently needed. The key to developing positive working conditions is to recruit more staff, improve wages, and invest in ongoing training and development opportunities. Ensuring sufficient staff-to-resident ratios is crucial to prevent staff turnover and provide quality care, as Slovenia and other Central and Eastern European countries were below the EU average in terms of staff-to-resident ratios [41]. It is also important to ensure that LTC staff have access to the resources and support they need to perform their roles effectively. It is very important to create a supportive working atmosphere where staff are adequately informed, listened to, and respected and where the hard work and commitment of the care staff are recognized, properly paid, and awarded. At the social level, it would also be important to ensure a better media representation of LTC work and workers and include geriatric content in all levels of education. First and foremost is addressing the problem of underinvestment in LTC.

### 4.4. Strengths and Limitations

This study contributes to filling the knowledge gap by highlighting the complex micro-, meso-, and macro-level factors and considerations associated with employment termination in LTC facilities.

The main limitation of the study is the use of the intention to quit one’s job. This study, like others, uses intentional turnover as an indicator of staff instability because actual turnover is difficult to analyze without longitudinal data. The statement “seriously considering quitting” must be regarded as an inaccurate indicator of actual turnover, even though turnover intention correlates with actual termination afterward. The study did not use an objective assessment of the working conditions but relied on LTC workers’ reports. However, these limitations also characterize previous studies of workers’ intentions to leave their jobs. 

Although we included micro, meso, and macro factors, the assessment tool was still limited. A standardized questionnaire was used to assess the work environment but not social-level factors, which include social recognition from social authorities, the media, and the public but not the local community, other sources, and actors (e.g., city leaders) who may also contribute to the (mis)recognition of work in LTC. Further research could include additional elements of social recognition.

The research found that more than half of LTC workers intended to quit their jobs and leave the LTC sector. On this basis, future research should investigate how many of these LTC workers actually left the job and what were the main factors that influenced and led them to leave the sector.

## 5. Conclusions

The results show that more than half of LTC workers intended to quit their jobs and leave the LTC sector. LTC workers who intended to leave were generally younger, had worked in the LTC sector for a shorter period of time, were mainly employed in the public sector, especially in nursing homes, and earned less. The connection between the intention to leave and the factors at the macro-, meso-, and micro-level is very high. Over 75% of the variance of intention to leave was explained by the linear influence of sociodemographic characteristics, social recognition, geriatric education, and work environment. At the micro level, among sociodemographic factors (education, income, shift work, dependence of household members, health status, and organizational form of the institution), income has the greatest influence on the intention to leave. At the meso level, the work environment has an important influence on the intention to leave. LTC workers are more likely to stay in the LTC sector if they feel safe, have more opportunities for training, have good relationships with patients or residents, do not feel overworked, believe that the quality of LTC in the organization is high, and receive more support from management, including adequate compensation for their work and contribution, i.e., paid overtime and with awards or honors. At the societal level, social recognition and geriatric education have an impact on the intention to leave work in LTC. Most respondents feel that society (social authorities, media, and the general public) does not value their work in LTC, in contrast to their family members and close friends who highly appreciate and value their work. Urgent measures for improving the work environment are needed.

## Figures and Tables

**Figure 1 ijerph-20-06612-f001:**
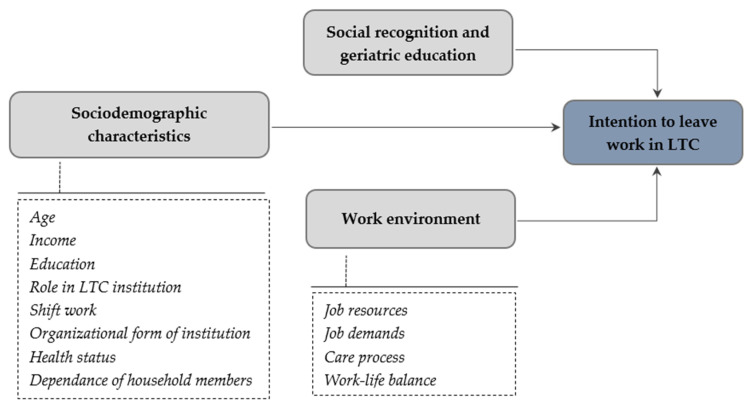
Conceptual model.

**Figure 2 ijerph-20-06612-f002:**
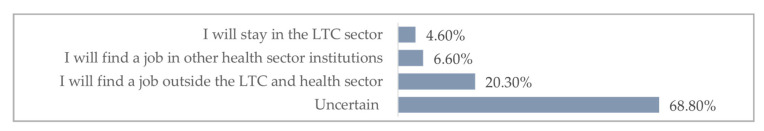
Area for finding a new job.

**Table 1 ijerph-20-06612-t001:** Respondents’ characteristics (*N* = 452) (%).

Attribute	Category	Share of Total Respondents (in %)
Gender	Male	9.8
	Female	90.2
	21–30	14.4
	31–40	23.2
Age	41–50	32.1
	51–60	26.5
	>60	3.8
Highest education level	Primary school	1.1
	National professional qualification	2.9
	Vocational school	8.8
	Secondary school	32.5
	College degree	26.8
	University undergraduate	20.4
	Master’s degree	7.1
	Doctoral degree	0.4
Organizational form of the institution	Public	85.0
	Private	15.0
Type of LTC organization	Nursing home	39.8
	Specialized social welfare institution	20.6
	Occupational activity center	7.5
	Center for training, work, and care	4.0
	Center for social work	20.8
	Home care assistant institution	4.4
	Other	2.9
Role in the LTC institution	Management	17.7
	Member of the nursing/care team (e.g., social worker, nurse)	45.6
	Home help manager/coordinator	7.1
	Member of the unit for strengthening and maintaining independence (e.g., kinesiologist, physiotherapist)	6.2
	Independent contractor (e.g., private visiting nurse, private personal assistant)	13.7
	Other (e.g., administrator, pedagogue, pharmacist)	9.7
Shift work	Single shift	48.2
	Second shift	31.7
	Third shift	20.1
Income (monthly net wage)	<EUR 1100	41.3
	EUR 1101–1700	38.8
	>EUR 1701	12.0
	I do not want to answer	8.0
Health status	I have chronic diseases	27.2
	I do not have chronic diseases	72.8
Dependent household member	Yes	67.5
	No	32.5

**Table 2 ijerph-20-06612-t002:** Differences in intention to leave by sociodemographic characteristics.

Variable [min–max]	Median	Value	Intention to Leave			*Kruskal-Wallis H*	*p*-Value
			No	Undefined	Yes		
Age [2,3,4,5,6]	4	>Median	93	37	7	38,632	<0.000 ***
		<=Median	122	126	67		
Years of service in LTC [1,2,3,4,5,6,7]	3	>Median	113	89	19	1318	<0.000 ***
		<=Median	102	74	53		
Organizational form of the institution [1,2,3]	1	>Median	29	19	20	10,503	0.005 **
		<=Median	186	144	54		
Type of organization [1,2,3,4,5,6,7,8]	2	>Median	109	54	16	27,867	<0.000 ***
		<=Median	106	109	58		
Income [1,2,3,4,5,6,7,8]	3	>Median	96	66	16	9226	0.010 **
		<=Median	119	96	58		
Dependent household members [1,2]	1	>Median	86	41	20	10,506	0.005 **
		<=Median	129	122	54		
Social recognition of LTC workers [1,2,3,4,5]	2	>Median	119	56	22	28,803	<0.000 ***
		<=Median	96	107	52		

** *p* < 0.01, *** *p* < 0.001.

**Table 3 ijerph-20-06612-t003:** The influence of underlying dimensions on the intention to leave work in LTC.

Variable	Model	*B*	Coefficients Std. Error	*t*	*p*-Value
	(Constant)	−2.183	0.960	−2274	0.030 *
Sociodemographic characteristics	Education	0.375	0.064	5875	<0.000 ***
	Income	0.641	0.351	1827	0.047 *
	Shift work	0.214	0.055	3920	<0.000 ***
	Dependent household member	0.291	0.153	1896	0.047 *
	Health status	0.322	0.195	6774	<0.000 ***
	Organizational form of the institution	0.331	0.203	6556	<0.000 ***
Social recognition and geriatric education	Social recognition of LTC workers	−0.419	0.083	−5075	<0.000 ***
	Social recognition of LTC work	−0.597	0.158	3773	<0.001 ***
	Geriatric secondary and higher education	1.442	0.277	5202	<0.000 ***
Work environment (job resources)	Technical support	−0.731	0.369	−1981	0.049 *
	Wage supplements	−1.204	0.192	−6260	<0.000 ***
	Awards and honors	−1.279	0.216	−5912	<0.000 ***
	Flexibility of work	0.739	0.231	3197	<0.000 ***
	Stability of employment	−1.965	0.299	−6581	<0.000 ***
	Possibility of workplace training	−1.595	0.376	−4245	<0.000 ***
	Investing in employee development	−0.201	0.105	−1924	0.044 *
	Management’s support	−1.181	0.192	−6140	<0.000 ***
	Job autonomy	0.309	0.133	2328	0.027 *
Work environment (job demands)	Role overload	0.974	0.302	3228	0.003 **
	Workload	1.337	0.109	3109	0.004 **
	Interpersonal conflicts	−0.483	0.113	−4267	<0.000 ***
Work environment (care process)	Quality of care	1.144	0.255	4496	<0.000 ***
	Relationship with patients/residents	1.890	0.359	5269	<0.000 ***
Work environment (work-life balance)	Negative work-life balance	0.293	0.096	3047	0.005 **

* *p* < 0.05, ** *p* < 0.01, *** *p* < 0.001.

## Data Availability

The data presented in this study are available on request from the corresponding author.

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
