# Peer review of "Influence of Sociodemographic, Organizational, and Social Factors on Turnover Consideration Among Eldercare Workers: A Quantitative Survey"

_ijerph, 2023, doi:10.3390/ijerph20166612_

Round 1

Reviewer 1 Report

Dear authors, thank you for the opportunity to read your research.

The study is certainly relevant and of great practical value.

The introduction contains an analysis of studies on staff turnover in Staff turnover in long-term care. The purpose and hypothesis of the study are substantiated.

Materials and methods contain a description of the sample, the procedure for collecting material, a description of the survey and statistical methods for data analysis.

The results are described structurally and clearly, logically. The discussion of the results correlates the obtained results with the studies of other authors in detail, practical recommendations and limitations of the study are given.

At the same time, there are a number of recommendations to the article:

1. When describing the evaluation questionnaire to study the influence of meso-level or organizational factors on employee turnover, it is necessary to transfer information about checking its validity and reliability from paragraph 2.6 to 2.3.

To clearly demonstrate the reliability of the tool.

2. In section 2.6. it is only necessary to briefly indicate what statistical methods were used to assess the validity and reliability of the instrument. And then, separately and in detail, the methods used to test the hypothesis and achieve the goal of the study.

Now it is non-structurally presented.

3. The authors should indicate in paragraph 2.3., in connection with which this particular instrument was chosen, and not other existing ones (justify).

4. A small remark, as a rule, at p 0.000***, indicated in the table <0.001***

5. In the discussion of the results, there is no clear description regarding the research hypothesis, whether it was confirmed and in what part.

6. In the discussion of the results, it is possible to add an analysis of the existing links regarding the theories of staff motivation, which explain the mechanism for stimulating certain factors on employee job satisfaction.

7. The limitations of the study do not indicate limitations related to the assessment tool used.

8. Output of line 454-455 "The connection between intention to leave and the factors at the macro-, meso-, and micro-level is very high." it is necessary to specify, indicate which factors from these groups are of paramount importance.

After correcting these remarks, the article can be recommended for publication.

With best wishes and respect for your research, reviewer

English language and writing style is appropriate, understandable

Author Response

Dear reviewer,

We wish to thank you for your efforts, the time spent reviewing our manuscript, and constructive comments. The comments provided valuable insights that helped to refine the manuscript’s content. In the attached document, we try to address the issues raised as best as possible. 

We used track changes to make all the changes in the manuscript visible.

Once again, we thank you for the time spent reviewing our manuscript and hope that we have met your expectations. 

The authors

Reviewer 2 Report

  Influence of Sociodemographic, Organizational and Social Factors on Turnover Consideration among Eldercare Workers: a Quantitative Survey

Peer Reviewer 1 – Comments

Dear Authors,

     Here are my few comments to address. They are as follows:

1.   You mentioned existing studies but inserted only one citation (19). Include more than one reference [Line 58-60]

2.   Kindly rephrase that sentence to retain what you intend to say [Line 64-66]

3.   The correct word is cross-sectional correlation study; kindly revise. A cross-sectional correlation study could investigate whether exposure to certain factors, such as overeating, might correlate to particular outcomes, such as obesity. While this study cannot prove that overeating causes obesity, it can draw attention to a relationship worth investigating. [Line 131]

4.   Move this Line 135 to 154 to Results [Line 135-154]

5.   Method section should follow these steps as follows:

Methods:

Study design,

Study setting,

Sample size,

Sampling strategy including inclusion and exclusion criteria,

Variable measurement(s),

Data collection,

Ethical consideration, and

Data analysis  

6.   What is your outcome variable(s) and the explanatory variables. Kindly state them clearly and explain them in sequence [Line 152-178]

7.   Explain in-depth details the method of data collection [Line180-188]

8.   What type of regression analysis do you want to utilize? Is it Linear regression, Polynomial regression, Logistics regression, Quantile regression, Ridge regression, Lasso regression, Elastic Net regression or Principle components regression (PCR) [Line 217]

9.   Insert a Table for the demographic characteristics of this study's respondents

10.   Align your analysis with this study's objectives. Let the objectives follow in sequence to show the findings from the results.

11.   Let the discussion stem from the study results by objectives

12.   Separate the strengths and limitations of the study from the discussion section

13.   Where are the recommendations for this study

14.   Kindly submit this manuscript to a Professional English Editor to edit the entire manuscript.

Kindly submit this manuscript to a Professional English Editor to edit the entire manuscript.

Author Response

(The authors gave the same response as above.)
